# Neural Architecture Search on ImageNet in Four GPU Hours: A Theoretically Inspired Perspective

**Wuyang Chen, Xinyu Gong, Zhangyang Wang**
Department of Electrical and Computer Engineering
The University of Texas at Austin, Austin, TX, USA
`{wuyang.chen,xinyu.gong,atlaswang}@utexas.edu`

## Abstract

Neural Architecture Search (NAS) has been explosively studied to automate the discovery of top-performer neural networks. Current works require heavy training of supernet or intensive architecture evaluations, thus suffering from heavy resource consumption and often incurring search bias due to truncated training or approximations. Can we select the best neural architectures without involving any training and eliminate a drastic portion of the search cost? We provide an affirmative answer, by proposing a novel framework called *training-free neural architecture search* (**TE-NAS**). TE-NAS ranks architectures by analyzing the spectrum of the neural tangent kernel (NTK) and the number of linear regions in the input space. Both are motivated by recent theory advances in deep networks and can be computed without any training and any label. We show that: (1) these two measurements imply the *trainability* and *expressivity* of a neural network; (2) they strongly correlate with the network's test accuracy. Further on, we design a pruning-based NAS mechanism to achieve a more flexible and superior trade-off between the trainability and expressivity during the search. In NAS-Bench-201 and DARTS search spaces, TE-NAS completes high-quality search but only costs **0.5** and **4** GPU hours with one 1080Ti on CIFAR-10 and ImageNet, respectively. We hope our work inspires more attempts in bridging the theoretical findings of deep networks and practical impacts in real NAS applications. Code is available at: `https://github.com/VITA-Group/TENAS`.

## 1 Introduction

The recent development of deep networks significantly contributes to the success of computer vision. Thanks to many efforts by human designers, the performance of deep networks have been significantly boosted (Krizhevsky et al., 2012; Simonyan & Zisserman, 2014; Szegedy et al., 2015; He et al., 2016; Xie et al., 2017). However, the manual creation of new network architectures not only costs enormous time and resources due to trial-and-error, but also depends on the design experience that does not always scale up. To reduce the human efforts and costs, neural architecture search (**NAS**) has recently amassed explosive interests, leading to principled and automated discovery for good architectures in a given search space of candidates (Zoph & Le, 2016; Brock et al., 2017; Pham et al., 2018; Liu et al., 2018a; Chen et al., 2018; Bender et al., 2018; Gong et al., 2019; Chen et al., 2020a; Fu et al., 2020).

As an optimization problem, NAS faces two core questions: 1) "**how to evaluate**", i.e. the objective function that defines what are good architectures we want; 2) "**how to optimize**", i.e. by what means we could effectively optimize the objective function. These two questions are entangled and highly non-trivial, since the search spaces are of extremely high dimension, and the generalization ability of architectures cannot be easily inferred (Dong & Yang, 2020; Dong et al., 2020). Existing NAS methods mainly leverage the validation set and conduct accuracy-driven architecture optimization. They either formulate the search space as a super-network ("supernet") and make the training loss differentiable through the architecture parameters (Liu et al., 2018b), or treat the architecture selection as a sequential decision making process (Zoph & Le, 2016) or evolution of genetics (Real et al., 2019). However, these NAS algorithms suffer from heavy consumption of both time and GPU resources. Training a supernet till convergence is extremely slow, even with many effective heuristics for sampling or channel approximations (Dong & Yang, 2019; Xu et al., 2019). Approximated proxy

inference such as truncated training/early stopping can accelerate the search, but is well known to introduce search bias to the inaccurate results obtained (Pham et al., 2018; Liang et al., 2019; Tan et al., 2020). The heavy search cost not only slows down the discovery of novel architectures, but also blocks us from more meaningfully understanding the NAS behaviors.

On the other hand, the analysis of neural network's trainability (how effective a network can be optimized via gradient descent) and expressivity (how complex the function a network can represent) has witnessed exciting development recently in the deep learning theory fields. By formulating neural networks as a Gaussian Process (no training involved), the gradient descent training dynamics can be characterized by the Neural Tangent Kernel (NTK) of infinite (Lee et al., 2019) or finite (Yang, 2019) width networks, from which several useful measures can be derived to depict the network trainability at the initialization. Hanin & Rolnick (2019a;b); Xiong et al. (2020) describe another measure of network expressivity, also without any training, by counting the number of unique linear regions that a neural network can divide in its input space. We are therefore inspired to ask:

- ***How to optimize*** *NAS at network's initialization without involving any training, thus significantly eliminating a heavy portion of the search cost?*

- *Can we define **how to evaluate** in NAS by analyzing the trainability and expressivity of architectures, and further benefit our understanding of the search process?*

Our answers are **yes** to both questions. In this work, we propose TE-NAS, a framework for training-free neural architecture search. We leverage *two indicators*, the condition number of NTK and the number of linear regions, that can decouple and effectively characterize the trainability and expressivity of architectures respectively in complex NAS search spaces. Most importantly, these two indicators can be measured in a training-free and label-free manner, thus largely accelerates the NAS search process and benefits the understanding of discovered architectures. To our best knowledge, TE-NAS makes the first attempt to bridge the theoretical findings of deep neural networks and real-world NAS applications. While we intend not to claim that the two indicators we use are the only nor the best options, we hope our work opens a door to theoretically-inspired NAS and inspires the discovery of more deep network indicators. Our contributions are summarized as below:

- We identify and investigate two training-free and label-free indicators to rank the quality of deep architectures: the spectrum of their NTKs, and the number of linear regions in their input space. Our study finds that they reliably indicate the trainability and expressivity of a deep network respectively, and are strongly correlated with the network's test accuracy.

- We leverage the above two theoretically-inspired indicators to establish a training-free NAS framework, **TE-NAS**, therefore eliminating a drastic portion of the search cost. We further introduce a pruning-based mechanism, to boost search efficiency and to more flexibly trade-off between trainability and expressivity.

- In NAS-Bench-201/DARTS search spaces, **TE-NAS** discovers architectures with a strong performance at remarkably lower search costs, compared to previous efforts. With just one 1080Ti, it only costs 0.5 GPU hours to search on CIFAR10, and 4 GPU hours on ImageNet, respectively, setting the new record for ultra-efficient yet high-quality NAS.

## 2 RELATED WORKS

**Neural architecture search (NAS)** is recently proposed to accelerate the principled and automated discovery of high-performance networks. However, most works suffer from heavy search cost, for both weight-sharing based methods (Liu et al., 2018b; Dong & Yang, 2019; Liu et al., 2019; Yu et al., 2020a; Li et al., 2020a; Yang et al., 2020a) and single-path sampling-based methods (Pham et al., 2018; Guo et al., 2019; Real et al., 2019; Tan et al., 2020; Li et al., 2020c; Yang et al., 2020b). A one-shot super network can share its parameters to sampled sub-networks and accelerate the architecture evaluations, but it is very heavy and hard to optimize and suffers from a poor correlation between its accuracy and those of the sub-networks (Yu et al., 2020c). Sampling-based methods achieve more accurate architecture evaluations, but their truncated training still imposes bias to the performance ranking since this is based on the results of early training stages.

Instead of estimating architecture performance by direct training, people also try to predict network's accuracy (or ranking), called **predictor based NAS** methods (Liu et al., 2018a; Luo et al., 2018; Dai et al., 2019; Luo et al., 2020). Graph neural network (GNN) is a popular choice as the predictor model (Wen et al., 2019; Chen et al., 2020b). Siems et al. (2020) even propose the first large-scale

surrogate benchmark, where most of the architectures' accuracies are predicted by a pretrained GNN predictor. The learned predictor can achieve highly accurate performance evaluation. However, the data collection step - sampling representative architectures and train them till converge - still requires extremely high cost. People have to sample and train 2,000 to 50,000 architectures to serve as the training data for the predictor. Moreover, none of these works can demonstrate the cross-space transferability of their predictors. This means one has to repeat the data collection and predictor training whenever facing an unseen search space, which is highly nonscalable.

The heavy cost of architecture evaluation hinders the **understanding** of the NAS search process. Recent pioneer works like Shu et al. (2019) observed that DARTS and ENAS tend to favor architectures with wide and shallow cell structures due to their smooth loss landscape. Siems et al. (2020) studied the distribution of test error for different cell depths and numbers of parameter-free operators. Chen & Hsieh (2020) for the first time regularizes the Hessian norm of the validation loss and visualizes the smoother loss landscape of the supernet. Li et al. (2020b) proposed to approximate the validation loss landscape by learning a mapping from neural architectures to their corresponding validate losses. Still, these analyses cannot be directly leveraged to guide the design of network architectures.

Mellor et al. (2020) recently proposed a NAS framework that does not involve training, which shares the same motivation with us towards training-free architecture search at initialization. They empirically find that the correlation between sample-wise input-output Jacobian can indicate the architecture's test performance. However, why does the Jacobian work is not well explained and demonstrated. Their search performance on NAS-Bench-201 is still left behind by the state-of-the-art NAS works, and they did not extend to DARTs space.

Meanwhile, we see the evolving development of **deep learning theory** on neural networks. NTK (neural tangent kernel) is proposed to characterize the gradient descent training dynamics of infinite wide (Jacot et al., 2018) or finite wide deep networks (Hanin & Nica, 2019). Wide networks are also proved to evolve as linear models under gradient descent (Lee et al., 2019). This is further leveraged to decouple the trainability and generalization of networks (Xiao et al., 2019). Besides, a natural measure of ReLU network's expressivity is the number of linear regions it can separate in its input space (Raghu et al., 2017; Montúfar, 2017; Serra et al., 2018; Hanin & Rolnick, 2019a;b; Xiong et al., 2020). In our work, we for the first time discover two important indicators that can effectively rank architectures, thus bridging the theoretic findings and real-world NAS applications. Instead of claiming the two indicators we discover are the best, we believe there are more meaningful properties of deep networks that can benefit the architecture search process. We leave them as open questions and encourage the community to study.

## 3 METHODS

The core motivation of our TE-NAS framework is to achieve architecture evaluation without involving any training, to significantly accelerate the NAS search process and reduce the search cost. In section 3.1 we present our study on two important indicators that reflect the trainability and expressivity of a neural network, and in section 3.2 we design a novel pruning-based method that can achieve a superior trade-off between the two indicators.

### 3.1 ANALYZING TRAINABILITY AND EXPRESSIVITY OF DEEP NETWORKS

Trainability and expressivity are distinct notions regarding a neural network (Xiao et al., 2019). A network can achieve high performance only if the function it can represent is complex enough and at the same time, it can be effectively trained by gradient descent.

#### 3.1.1 TRAINABILITY BY CONDITION NUMBER OF NTK

The trainability of a neural network indicates how effective it can be optimized using gradient descent (Burkholz & Dubatovka, 2019; Hayou et al., 2019; Shin & Karniadakis, 2020). Although some heavy networks can theoretically represent complex functions, they not necessarily can be effectively trained by gradient descent. One typical example is that, even with a much more number of parameters, Vgg networks (Simonyan & Zisserman, 2014) usually perform worse and require more special engineering tricks compared with ResNet family (He et al., 2016), whose superior trainability property is studied by Yang & Schoenholz (2017).

Recent work (Jacot et al., 2018; Lee et al., 2019; Chizat et al., 2019) studied the gradient descent training of neural networks using a quantity called the neural tangent kernel (NTK). The finite width NTK is defined by $\hat{\Theta}(\boldsymbol{x}, \boldsymbol{x}') = J(\boldsymbol{x})J(\boldsymbol{x}')^T$, where $J_{i\alpha}(\boldsymbol{x}) = \partial_{\theta_\alpha} z_i^L(\boldsymbol{x})$ is the Jacobian evaluated at a point $\boldsymbol{x}$ for parameter $\theta_\alpha$, and $z_i^L$ is the output of the $i$-th neuron in the last output layer $L$.

Lee et al. (2019) further proves that wide neural networks evolve as linear models using gradient descent, and their training dynamics is controlled by ODEs that can be solved as

$$\mu_t(\boldsymbol{X}_{\text{train}}) = (\mathbf{I} - e^{-\eta\hat{\Theta}_{\text{train}}t})\boldsymbol{Y}_{\text{train}} \tag{1}$$

for training data. Here $\mu_t(\boldsymbol{x}) = \mathbb{E}[z_i^L(\boldsymbol{x})]$ is the expected outputs of an infinitely wide network. $\hat{\Theta}_{\text{train}}$ denotes the NTK between the training inputs, and $\boldsymbol{X}_{\text{train}}$ and $\boldsymbol{Y}_{\text{train}}$ are drawn from the training set $\mathcal{D}_{\text{train}}$. As the training step $t$ tends to infinity we can see that Eq. 1 reduce to $\mu(\boldsymbol{X}_{\text{train}}) = \boldsymbol{Y}_{\text{train}}$.

The relationship between the conditioning of $\hat{\Theta}$ and the trainability of networks is studied by Xiao et al. (2019), and we brief the conclusion as below. We can write Eq. 1 in terms of the spectrum of $\Theta$:

$$\mu_t(\boldsymbol{X}_{\text{train}})_i = (\mathbf{I} - e^{-\eta\lambda_i t})\boldsymbol{Y}_{\text{train},i}, \tag{2}$$

where $\lambda_i$ are the eigenvalues of $\hat{\Theta}_{\text{train}}$ and we order the eigenvalues $\lambda_0 \geq \cdots \geq \lambda_m$. As it has been hypothesized by Lee et al. (2019) that the maximum feasible learning rate scales as $\eta \sim 2/\lambda_0$, plugging this scaling for $\eta$ into Eq. 2 we see that the $\lambda_m$ will converge exponentially at a rate given by $1/\kappa$, where $\kappa = \lambda_0/\lambda_m$ is the condition number. Then we can conclude that if the $\kappa$ of the NTK associated with a neural network diverges then it will become untrainable, so we use $\kappa$ as a metric for trainability:

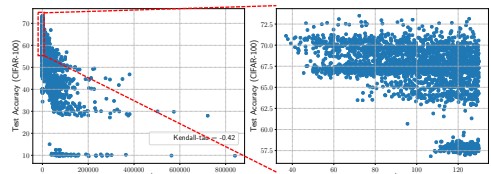

**Figure 1:** Condition number of NTK $\kappa_\mathcal{N}$ exhibits negative correlation with the test accuracy of architectures in NAS-Bench201 (Dong & Yang, 2020).

$$\kappa_\mathcal{N} = \frac{\lambda_0}{\lambda_m}. \tag{3}$$

$\kappa_\mathcal{N}$ is calculated without any gradient descent or label. Figure 1 demonstrates that the $\kappa_\mathcal{N}$ is negatively correlated with the architecture's test accuracy, with the Kendall-tau correlation as $-0.42$. Therefore, minimizing the $\kappa_\mathcal{N}$ during the search will encourage the discovery of architectures with high performance.

### 3.1.2 Expressivity by Number of Linear Regions

The expressivity of a neural network indicates how complex the function it can represent (Hornik et al., 1989; Giryes et al., 2016). For ReLU networks, each ReLU function defines a linear boundary and divides its input space into two regions. Since the composition of piecewise linear functions is still piecewise linear, every ReLU network can be seen as a piecewise linear function. The input space of a ReLU network can be partitioned into distinct pieces (i.e. linear regions) (Figure 2), each of which is associated with a set of affine parameters, and the function represented by the network is affine when restricted to each piece. Therefore, it is natural to measure the expressivity of a ReLU network with the number of linear regions it can separate.

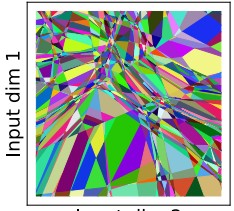

**Figure 2:** Example of linear regions divided by a ReLU network[1]

Following Raghu et al. (2017); Montúfar (2017); Serra et al. (2018); Hanin & Rolnick (2019a;b); Xiong et al. (2020), we first introduce the following definition of activation patterns and linear regions for ReLU CNNs.

**Definition 1. Activation Patterns and Linear Regions (Xiong et al. (2020))** *Let $\mathcal{N}$ be a ReLU CNN. An activation pattern of $\mathcal{N}$ is a function $\boldsymbol{P}$ from the set of neurons to $\{1, -1\}$, i.e., for each neuron $z$ in $\mathcal{N}$, we have $\boldsymbol{P}(z) \in \{1, -1\}$. Let $\theta$ be a fixed set of parameters (weights and biases) in $\mathcal{N}$, and $\boldsymbol{P}$ be an activation pattern. The region corresponding to $\boldsymbol{P}$ and $\theta$ is*

$$\boldsymbol{R}(\boldsymbol{P}; \theta) := \{\boldsymbol{x}^0 \in \mathbb{R}^{C \times H \times W} : z(\boldsymbol{x}^0; \theta) \cdot \boldsymbol{P}(z) > 0, \quad \forall z \in \mathcal{N}\}, \tag{4}$$

*where $z(\boldsymbol{x}^0; \theta)$ is the pre-activation of a neuron $z$. Let $R_{\mathcal{N},\theta}$ denote the number of linear regions of $\mathcal{N}$ at $\theta$, i.e., $R_{\mathcal{N},\theta} := \#\{\boldsymbol{R}(\boldsymbol{P}; \theta) : \boldsymbol{R}(\boldsymbol{P}; \theta) \neq \emptyset$ for some activation pattern $\boldsymbol{P}\}$.*

---

[1]Plot is generated by us with the same method described by Hanin & Rolnick (2019a).

Eq. 4 tells us that a linear region in the input space is a set of input data $x^0$ that satisfies a certain fixed activation pattern $P(z)$, and therefore the number of linear regions $R_{\mathcal{N},\theta}$ measures how many unique activation patterns that can be divided by the network.

In our work, we repeat the measurement of the number of linear regions by sampling network parameters from the Kaiming Norm Initialization (He et al., 2015), and calculate the average as the approximation to its expectation:

$$\hat{R}_{\mathcal{N}} \simeq \mathbb{E}_{\theta} R_{\mathcal{N},\theta} \qquad (5)$$

We iterate through all architectures in NAS-Bench-201 (Dong & Yang, 2020), and calculate their numbers of linear regions (without any gradient descent or label). Figure 3 demonstrates that the number of linear regions is positively correlated with the architecture's test accuracy, with the Kendall-tau correlation as 0.5. Therefore, maximizing the number of linear regions during the search will also encourage the discovery of architectures with high performance.

Finally, in Figure 4 we analyze the operator composition of top 10% architecture by maximizing $\hat{R}_{\mathcal{N}}$ and minimizing $\kappa_{\mathcal{N}}$, respectively. We can clearly see that

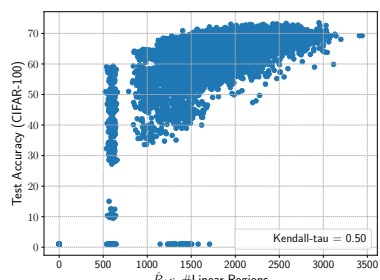

**Figure 3:** Number of linear regions $\hat{R}_{\mathcal{N}}$ of architectures in NAS-Bench201 exhibits positive correlation with test accuracies.

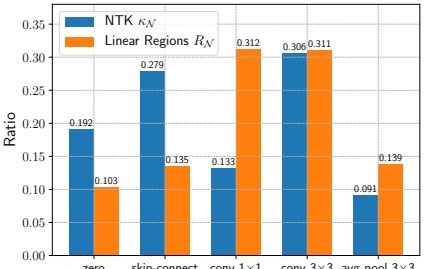

**Figure 4:** $\kappa_{\mathcal{N}}$ and $\hat{R}_{\mathcal{N}}$ prefer different operators in NAS-Bench201.

$\hat{R}_{\mathcal{N}}$ and $\kappa_{\mathcal{N}}$ have different preferences for choosing operators. They both choose a large ratio of conv$3 \times 3$ for high generalization performance. But meanwhile, $\hat{R}_{\mathcal{N}}$ heavily selects conv$1 \times 1$, and $\kappa_{\mathcal{N}}$ leads to skip-connect, favoring the gradient flow.

### 3.2 PRUNING-BY-IMPORTANCE ARCHITECTURE SEARCH

Given the strong correlation between the architecture's test accuracy and its $\kappa_{\mathcal{N}}$ and $\hat{R}_{\mathcal{N}}$, how to build an efficient NAS framework on top of them? We motivate this section by addressing two questions:

*1. How to combine $\kappa_{\mathcal{N}}$ and $\hat{R}_{\mathcal{N}}$ together, with a good explicit trade-off?*

We first need to turn the two measurements $\kappa_{\mathcal{N}}$ and $\hat{R}_{\mathcal{N}}$ into one combined function, based on which we can rank architectures. As seen in Figure 1 and 3, the magnitudes of $\kappa_{\mathcal{N}}$ and $\hat{R}_{\mathcal{N}}$ differ much. To avoid one overwhelming the other numerically, one possible remedy is normalization; but we cannot pre-know the ranges nor the value distributions of $\kappa_{\mathcal{N}}$ and $\hat{R}_{\mathcal{N}}$, before computing them over a search space. In order to make our combined function *well defined before search* and *agnostic to the search space*, instead of using the numerical values of $\kappa_{\mathcal{N}}$ and $\hat{R}_{\mathcal{N}}$, we could refer to their relative rankings. Specifically, each time by comparing the sampled set of architectures peer-to-peer, we can directly sum up the two relative rankings of $\kappa_{\mathcal{N}}$ and $\hat{R}_{\mathcal{N}}$ as the selection criterion. The equal-weight summation treats trainability and expressivity with the same importance conceptually[1] and delivers the best empirical result: we thus choose it as our default combined function. We also tried some other means to combine the two, and the ablation studies can be found in Appendix D.2.

*2. How to search more efficiently?*

Sampling-based methods like reinforcement learning or evolution can use rankings as the reward or filtering metric. However, they are inefficient, especially for complex cell-based search space. Consider a network stacked by repeated cells (directed acyclic graphs) (Zoph et al., 2018; Liu et al., 2018b). Each cell has $E$ edges, and on each edge we only select one operator out of $|\mathcal{O}|$ ($\mathcal{O}$ is the set of operator candidates). There are $|\mathcal{O}|^E$ unique cells, and for sampling-based methods, $\gamma \cdot |\mathcal{O}|^E$ networks have to be sampled during the search. The ratio $\gamma$ can be interpreted as the sampling efficiency: a method with small $\gamma$ can find good architectures faster. However, the search time cost of sampling-based methods still scales up with the size of the search space, i.e., $|\mathcal{O}|^E$.

---

[1] We tried some weighted summations of the two, and find their equal-weight summation to perform the best.

Inspired by recent works on pruning-from-scratch (Lee et al., 2018; Wang et al., 2020), we propose a pruning-by-importance NAS mechanism to quickly shrink the search possibilities and boost the efficiency further, reducing the cost from $|\mathcal{O}|^E$ to $|\mathcal{O}| \cdot E$. Specifically, we start the search with a super-network $\mathcal{N}_0$ composed of all possible operators and edges. In the outer loop, for every round we prune one operator on each edge. The outer-loop stops when the current supernet $\mathcal{N}_t$ is a single-path network[2], i.e., the algorithm will return us the final searched architecture. For the inner-loop, we measure the change of $\kappa_{\mathcal{N}}$ and $\hat{R}_{\mathcal{N}}$ before and after pruning each individual operator, and assess its importance using the sum of two ranks. We order all currently available operators in terms of their importance, and prune the lowest-importance operator on each edge.

The whole pruning process is extremely fast. As we will demonstrate later, our approach is principled and can be applied to different spaces without making any modifications. This pruning-by-importance mechanism may also be extended to indicators beyond $\kappa_{\mathcal{N}}$ and $\hat{R}_{\mathcal{N}}$. We summarize our training-free and pruning-based NAS framework, TE-NAS, in Algorithm 1.

---

**Algorithm 1:** TE-NAS: Training-free Pruning-based NAS via Ranking of $\kappa_{\mathcal{N}}$ and $\hat{R}_{\mathcal{N}}$.

---

1  **Input:** supernet $\mathcal{N}_0$ stacked by cells, each cell has $E$ edges, each edge has $|\mathcal{O}|$ operators, step $t = 0$.
2  **while** $\mathcal{N}_t$ *is not a single-path network* **do**
3      **for** *each operator $o_j$ in $\mathcal{N}_t$* **do**
4          $\Delta\kappa_{t,o_j} = \kappa_{\mathcal{N}_t} - \kappa_{\mathcal{N}_t \backslash o_j}$          ▷ the higher $\Delta\kappa_{t,o_j}$ the more likely we will prune $o_j$
5          $\Delta R_{t,o_j} = R_{\mathcal{N}_t} - R_{\mathcal{N}_t \backslash o_j}$          ▷ the lower $\Delta R_{t,o_j}$ the more likely we will prune $o_j$
6      Get importance by $\kappa_{\mathcal{N}}$: $s_\kappa(o_j) =$ index of $o_j$ in descendingly sorted list $[\Delta\kappa_{t,o_1}, ..., \Delta\kappa_{t,o_{|\mathcal{N}_t|}}]$
7      Get importance by $R_{\mathcal{N}}$: $s_R(o_j) =$ index of $o_j$ in ascendingly sorted list $[\Delta R_{t,o_1}, ..., \Delta R_{t,o_{|\mathcal{N}_t|}}]$
8      Get importance $s(o_j) = s_\kappa(o_j) + s_R(o_j)$
9      $\mathcal{N}_{t+1} = \mathcal{N}_t$
10     **for** *each edge $e_i$, $i = 1, ..., E$* **do**
11         $j^* = \arg\min_j \{s(o_j) : o_j \in e_i\}$      ▷ find the operator with greatest importance on each edge.
12         $\mathcal{N}_{t+1} = \mathcal{N}_{t+1} \backslash o_{j^*}$
13     $t = t + 1$
14 **return** *Pruned single-path network $\mathcal{N}_t$.*

---

### 3.2.1 VISUALIZATION OF SEARCH PROCESS

TE-NAS benefits us towards a better understanding of the search process. We can analyze the trajectory of $\kappa_{\mathcal{N}}$ and $\hat{R}_{\mathcal{N}}$ during the search. It is worth noting that our starting point $\mathcal{N}_0$, the un-pruned supernet, is assumed to be of the highest expressivity (as it is composed of all operators in the search space and has the largest number of parameters and ReLU functions). However, it has poor trainability, as people find many engineering techniques are required to effectively training the supernet (Yu et al., 2020a;b). Therefore, during pruning we are expecting to strengthen the trainability of the supernet, while retaining its expressivity as much as possible.

As we observe in Figure 5, the supernet $\mathcal{N}$ is first pruned by quickly reducing $\kappa_{\mathcal{N}}$, i.e., increasing the network's trainability. After that, as the improvement of $\kappa_{\mathcal{N}}$ is almost plateaued, the method carefully fine-tunes the architecture without sacrificing too much expressivity $\hat{R}_{\mathcal{N}}$.

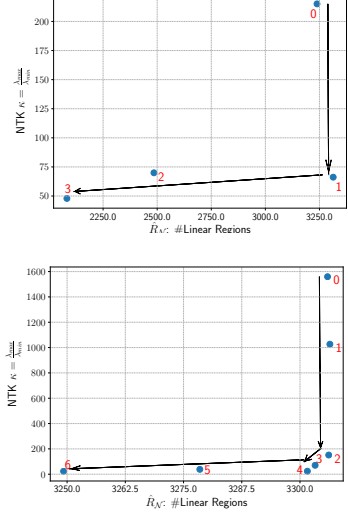

**Figure 5:** Pruning trajectory on NAS-Bench-201 (top) and DARTs search space (bottom). Number "0" indicates the supernet $\mathcal{N}_0$ before any pruning, which is of high expressivity but poor trainability.

---

[2]Different search spaces may have different criteria for the single-path network. In NAS-Bench201 (Dong & Yang, 2020) each edge only keeps one operator at the end of the search, while in DARTS space (Liu et al., 2018b) there are two operators on each edge in the searched network.

## 4 EXPERIMENTS

In this section, we evaluate our TE-NAS on two search spaces: NAS-Bench-201 (Dong & Yang, 2020) and DARTS (Liu et al., 2018b). Search and training protocols are summarized in Appendix A. Our code is available at: `https://github.com/VITA-Group/TENAS`.

### 4.1 RESULTS ON NAS-BENCH-201

NAS-Bench-201 (Dong & Yang, 2020) provides a standard cell-based search space (containing 15,625 architectures) and a database of architecture's performance evaluated under a unified protocol. The network's test accuracy can be directly obtained by querying the database, which facilitates people to focus on studying NAS algorithms without network evaluation. NAS-Bench-201 supports three datasets (CIFAR-10, CIFAR-100, ImageNet-16-120 (Chrabaszcz et al., 2017)). The operation space contains *none (zero)*, *skip connection*, *conv*$1 \times 1$, *conv*$3 \times 3$ *convolution*, and *average pooling* $3 \times 3$. We refer to their paper for details of the space. Our search is dataset-specific, i.e. the search and evaluation are conducted on the same dataset.

**Table 1:** Comparison with state-of-the-art NAS methods on NAS-Bench-201. Test accuracy with mean and deviation are reported. "optimal" indicates the best test accuracy achievable in NAS-Bench-201 search space.

| Architecture | CIFAR-10 | CIFAR-100 | ImageNet-16-120 | Search Cost (GPU sec.) | Search Method |
|---|---|---|---|---|---|
| ResNet (He et al., 2016) | 93.97 | 70.86 | 43.63 | - | - |
| RSPS (Li & Talwalkar, 2020) | 87.66(1.69) | 58.33(4.34) | 31.14(3.88) | 8007.13 | random |
| ENAS (Pham et al., 2018) | 54.30(0.00) | 15.61(0.00) | 16.32(0.00) | 13314.51 | RL |
| DARTS (1st) (Liu et al., 2018b) | 54.30(0.00) | 15.61(0.00) | 16.32(0.00) | 10889.87 | gradient |
| DARTS (2nd) (Liu et al., 2018b) | 54.30(0.00) | 15.61(0.00) | 16.32(0.00) | 29901.67 | gradient |
| GDAS (Dong & Yang, 2019) | 93.61(0.09) | 70.70(0.30) | 41.84(0.90) | 28925.91 | gradient |
| NAS w.o. Training (Mellor et al., 2020) | 91.78(1.45) | 67.05(2.89) | 37.07(6.39) | 4.8 | training-free |
| TE-NAS (ours) | **93.9(0.47)** | **71.24(0.56)** | **42.38(0.46)** | 1558 | training-free |
| **Optimal** | 94.37 | 73.51 | 47.31 | - | - |

We run TE-NAS for four independent times with different random seeds, and report the mean and standard deviation in Table 1. We can see that TE-NAS achieves the best accuracy on all three datasets, and largely reduces the search cost ($5\times \sim 19\times$ reduction). Although Mellor et al. (2020) requires even less search time (by only sampling 25 architectures), they suffer from much inferior accuracy performance, with notably larger deviations across different search rounds.

### 4.2 RESULTS ON CIFAR-10 WITH DARTS SEARCH SPACE

**Architecture Space** The DARTs operation space $\mathcal{O}$ contains eight choices: *none (zero)*, *skip connection*, *separable convolution* $3 \times 3$ and $5 \times 5$, *dilated separable convolution* $3 \times 3$ and $5 \times 5$, *max pooling* $3 \times 3$, *average pooling* $3 \times 3$. Following previous works (Liu et al., 2018b; Chen et al., 2019; Xu et al., 2019), for evaluation phases, we stack 20 cells to compose the network and set the

**Table 2:** Comparison with state-of-the-art NAS methods on CIFAR-10.

| Architecture | Test Error (%) | Params (M) | Search Cost (GPU days) | Search Method |
|---|---|---|---|---|
| AmoebaNet-A (Real et al., 2019) | 3.34(0.06) | 3.2 | 3150 | evolution |
| PNAS (Liu et al., 2018a)[*] | 3.41(0.09) | 3.2 | 225 | SMBO |
| ENAS (Pham et al., 2018) | 2.89 | 4.6 | 0.5 | RL |
| NASNet-A (Zoph et al., 2018) | 2.65 | 3.3 | 2000 | RL |
| DARTS (1st) (Liu et al., 2018b) | 3.00(0.14) | 3.3 | 0.4 | gradient |
| DARTS (2nd) (Liu et al., 2018b) | 2.76(0.09) | 3.3 | 1.0 | gradient |
| SNAS (Xie et al., 2018) | 2.85(0.02) | 2.8 | 1.5 | gradient |
| GDAS (Dong & Yang, 2019) | 2.82 | 2.5 | 0.17 | gradient |
| BayesNAS (Zhou et al., 2019) | 2.81(0.04) | 3.4 | 0.2 | gradient |
| ProxylessNAS (Cai et al., 2018)[†] | 2.08 | 5.7 | 4.0 | gradient |
| P-DARTS (Chen et al., 2019) | 2.50 | 3.4 | 0.3 | gradient |
| PC-DARTS (Xu et al., 2019) | 2.57(0.07) | 3.6 | 0.1 | gradient |
| SDARTS-ADV (Chen & Hsieh, 2020) | 2.61(0.02) | 3.3 | 1.3 | gradient |
| TE-NAS (ours) | 2.63(0.064) | 3.8 | 0.05[‡] | training-free |

[*] No cutout augmentation.

[†] Different space: PyramidNet (Han et al., 2017) as the backbone.

[‡] Recorded on a single GTX 1080Ti GPU.

initial channel number as 36. We place the reduction cells at the 1/3 and 2/3 of the network and each cell consists of six nodes.

**Results** We run TE-NAS for four independent times with different random seeds, and report the mean and standard deviation. Table 2 summarizes the performance of TE-NAS compared with other popular NAS methods. TE-NAS achieves a test error of 2.63%, ranking among the top of recent NAS results, but meanwhile largely reduces the search cost to only 0.05 GPU-day. ProxylessNAS achieves the lowest test error, but it searches on a different space with a much longer search time and has a larger model size. Besides, Mellor et al. (2020) did not extend to their Jacobian-based framework to DARTs search space for CIFAR-10 or ImageNet classification.

## 4.3 RESULTS ON IMAGENET WITH DARTS SEARCH SPACE

**Architecture Space** Following previous works (Xu et al., 2019; Chen et al., 2019), the architecture for ImageNet is slightly different from that for CIFAR-10. During retraining evaluation, the network is stacked with 14 cells with the initial channel number set to 48, and we follow the mobile setting to control the FLOPs not exceed 600 MB by adjusting the channel number. The spatial resolution is downscaled from $224 \times 224$ to $28 \times 28$ with the first three convolution layers of stride 2.

**Results** As shown in Table 3, we achieve a top-1/5 test error of 24.5%/7.5%, achieving competitive performance with recent state-of-the-art works in the ImageNet mobile setting. However, TE-NAS only cost four GPU hours with only one 1080Ti. Searching on ImageNet takes a longer time than on CIFAR-10 due to the larger input size and more network parameters.

**Table 3:** Comparison with state-of-the-art NAS methods on ImageNet under the mobile setting.

| Architecture | Test Error(%) | | Params (M) | Search Cost (GPU days) | Search Method |
|---|---|---|---|---|---|
| | top-1 | top-5 | | | |
| NASNet-A (Zoph et al., 2018) | 26.0 | 8.4 | 5.3 | 2000 | RL |
| AmoebaNet-C (Real et al., 2019) | 24.3 | 7.6 | 6.4 | 3150 | evolution |
| PNAS (Liu et al., 2018a) | 25.8 | 8.1 | 5.1 | 225 | SMBO |
| MnasNet-92 (Tan et al., 2019) | 25.2 | 8.0 | 4.4 | - | RL |
| DARTS (2nd) (Liu et al., 2018b) | 26.7 | 8.7 | 4.7 | 4.0 | gradient |
| SNAS (mild) (Xie et al., 2018) | 27.3 | 9.2 | 4.3 | 1.5 | gradient |
| GDAS (Dong & Yang, 2019) | 26.0 | 8.5 | 5.3 | 0.21 | gradient |
| BayesNAS (Zhou et al., 2019) | 26.5 | 8.9 | 3.9 | 0.2 | gradient |
| P-DARTS (CIFAR-10) (Chen et al., 2019) | 24.4 | 7.4 | 4.9 | 0.3 | gradient |
| P-DARTS (CIFAR-100) (Chen et al., 2019) | 24.7 | 7.5 | 5.1 | 0.3 | gradient |
| PC-DARTS (CIFAR-10) (Xu et al., 2019) | 25.1 | 7.8 | 5.3 | 0.1 | gradient |
| TE-NAS (ours) | 26.2 | 8.3 | 6.3 | 0.05 | training-free |
| PC-DARTS (ImageNet) (Xu et al., 2019)[†] | 24.2 | 7.3 | 5.3 | 3.8 | gradient |
| ProxylessNAS (GPU) (Cai et al., 2018)[†] | 24.9 | 7.5 | 7.1 | 8.3 | gradient |
| TE-NAS (ours)[†] | 24.5 | 7.5 | 5.4 | 0.17 | training-free |

[†] The architecture is searched on ImageNet, otherwise it is searched on CIFAR-10 or CIFAR-100.

## 5 CONCLUSION

The key questions in Neural Architecture Search (NAS) are "what are good architectures" and "how to find them". Validation loss or accuracy are possible answers but not enough, due to their search bias and heavy evaluation cost. Our work demonstrates that two theoretically inspired indicators, the spectrum of NTK and the number of linear regions, not only strongly correlate with the network's performance, but also benefit the reduced search cost and decoupled analysis of the network's trainability and expressivity. Without involving any training, our TE-NAS achieve competitive NAS performance with minimum search time. We for the first time bridge the gap between the theoretic findings of deep neural networks and real-world NAS applications, and we encourage the community to further explore more meaningful network properties so that we will have a better understanding of good architectures and how to search them.

## ACKNOWLEDGEMENT

This work is supported in part by the NSF Real-Time Machine Learning program (Award Number: 2053279), and the US Army Research Office Young Investigator Award (W911NF2010240).

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

## A    IMPLEMENTATION DETAILS

For $\kappa_{\mathcal{N}}$ we sample one mini-batch of size 32 from the training set, and calculate $\hat{\Theta}(\boldsymbol{x}, \boldsymbol{x}') = J(\boldsymbol{x})J(\boldsymbol{x}')^T$. For $\hat{R}_{\mathcal{N}}$ we sample 5000 images, forward them through the network, and collect the activation patterns from all ReLU layers. The calculation of both $\kappa_{\mathcal{N}}$ and $\hat{R}_{\mathcal{N}}$ are repeated three times in all experiments, where each time the network weights are randomly drawn from Kaiming Norm Initialization (He et al., 2015) without involving any training (network weights are fixed).

Our retraining settings (after search) follow previous works (Xu et al., 2019; Chen et al., 2019; Chen & Hsieh, 2020). On CIFAR-10, we train the searched network with cutout regularization of length 16, drop-path (Zoph et al., 2018) with probability as 0.3, and an auxiliary tower of weight 0.4. On ImageNet, we also use label smoothing during training. On both CIFAR-10 and ImageNet, the network is optimized by an SGD optimizer with cosine annealing, with learning rate initialized as 0.025 and 0.5, respectively.

## B    SEARCHED ARCHITECTURE

We visualize the searched normal and reduction cells in figure 6 and 7, which is directly searched on CIFAR-10 and ImageNet respectively.

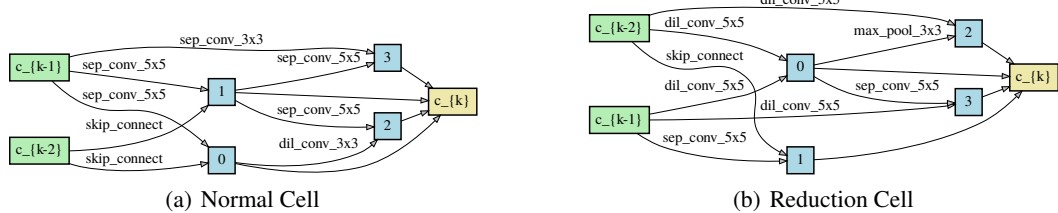

|           (a) Normal Cell           |           (b) Reduction Cell           |

**Figure 6:** Normal and Reduction cells discovered by TE-NAS on CIFAR-10.

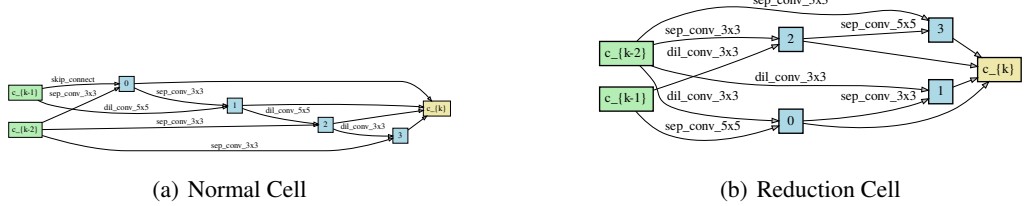

|           (a) Normal Cell           |           (b) Reduction Cell           |

**Figure 7:** Normal and Reduction cells discovered by TE-NAS on imageNet.

## C    DEPTH AND WIDTH PREFERENCE OF $\kappa_{\mathcal{N}}$ AND $\hat{R}_{\mathcal{N}}$ IN DARTs SPACE

To analyze the impact of different architectures on trainability and expressivity in DARTs search space, we visualize $\kappa_{\mathcal{N}}$ and $\hat{R}_{\mathcal{N}}$ with different depths and width. Following Shu et al. (2019), the depth of a cell is defined as the number of connections on the longest path from input nodes to the output node, and the width of a cell is the summation of the edges of the intermediate nodes that are connected to the input nodes. We randomly sample 20,000 architectures in DARTs space, and plot the visualizations in Figure 8. Good architectures should exhibit low $\kappa_{\mathcal{N}}$ (good trainability, blue dots in Figure 8(a)) and high $\hat{R}_{\mathcal{N}}$ (powerful expressivity, red dots in Figure 8(b)). Therefore, $\kappa_{\mathcal{N}}$ and $\hat{R}_{\mathcal{N}}$ tell us that in DARTs space shallow but wide cells are preferred to favor both trainability and expressivity. This conclusion matches the findings by Shu et al. (2019): existing NAS algorithms tend to favor architectures with wide and shallow cell structures, which enjoy fast convergence with smooth loss landscape and accurate gradient information.

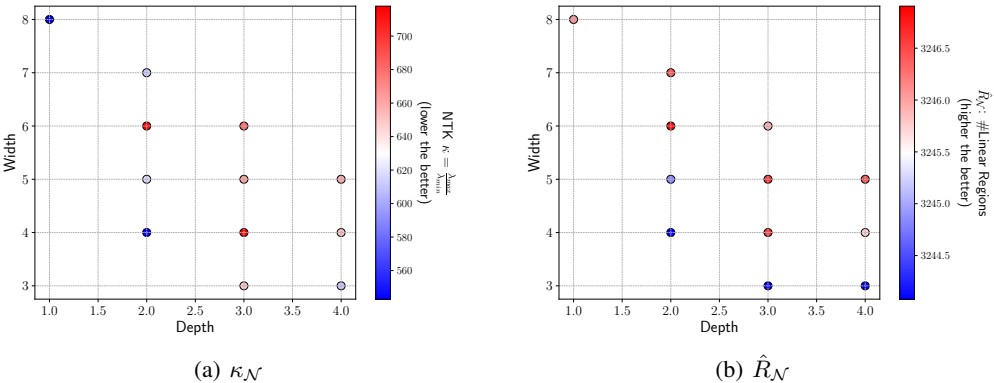

(a) $\kappa_{\mathcal{N}}$           (b) $\hat{R}_{\mathcal{N}}$

**Figure 8:** Depth and width preference of (a) $\kappa_{\mathcal{N}}$ and (b) $\hat{R}_{\mathcal{N}}$ on DARTs Search Space.

## D   MORE ABLATION STUDIES

### D.1   SEARCH WITH ONLY $\kappa_{\mathcal{N}}$ OR $\hat{R}_{\mathcal{N}}$

As we observed in Table 4, searching with only $\kappa_{\mathcal{N}}$ or $\hat{R}_{\mathcal{N}}$ leads to inferior performance, which indicates the importance of maintaining both trainability and expressivity during the search.

**Table 4:** Search with only $\kappa_{\mathcal{N}}$ or $\hat{R}_{\mathcal{N}}$ on CIFAR-100 in NAS-Bench-201 space.

| Methods | CIFAR-100 Test Accuracy |
|---|---|
| Prune with only $\kappa_{\mathcal{N}}$ | 69.25 (1.29) |
| Prune with only $\hat{R}_{\mathcal{N}}$ | 70.48 (0.29) |
| TE-NAS | **71.24** (0.56) |

### D.2   DIFFERENT COMBINATION OPTIONS FOR $\kappa_{\mathcal{N}}$ AND $\hat{R}_{\mathcal{N}}$

Pruning by $s(o_j) = s_\kappa(o_j) + s_R(o_j)$ is not the only option (see Algorithm 1). Here in this study we consider more:

1) pruning by $s(o_j) = \min(s_\kappa(o_j), s_R(o_j))$, i.e., pruning by the worst case.
2) Pruninng by $s(o_j) = \max(s_\kappa(o_j), s_R(o_j))$, i.e., pruning by the best case.
3) pruning by summation of changes $\Delta\kappa_{t,o_j} + \Delta R_{t,o_j}$, i.e., directly use the numerical values of the changes.

As we observed in Table 5, our TE-NAS stands out of all options. This means a good trade-off between $\kappa_{\mathcal{N}}$ and $\hat{R}_{\mathcal{N}}$ are important, and also the ranking strategy is better than directly using numerical values.

**Table 5:** Search with only $\kappa_{\mathcal{N}}$ or $\hat{R}_{\mathcal{N}}$ on CIFAR-100 in NAS-Bench-201 space.

| Methods | CIFAR-100 Test Accuracy |
|---|---|
| Prune by $s(o_j) = \min(s_\kappa(o_j), s_R(o_j))$ | 70.75 (0.73) |
| Prune by $s(o_j) = \max(s_\kappa(o_j), s_R(o_j))$ | 70.33 (1.09) |
| Prune by $\Delta\kappa_{t,o_j} + \Delta R_{t,o_j}$ | 70.47 (0.68) |
| Prune by $s(o_j) = s_\kappa(o_j) + s_R(o_j)$ (TE-NAS) | **71.24** (0.56) |

### D.3 Correlation between Test Accuracy and Combination of $\kappa_{\mathcal{N}}$ and $\hat{R}_{\mathcal{N}}$

Figure 9 indicates that by using the summation of the ranking of both $\kappa_{\mathcal{N}}$ and $\hat{R}_{\mathcal{N}}$, the combined metric achieves a much higher correlation with the test accuracy. The reason can be explained by Figure 4, as $\kappa_{\mathcal{N}}$ and $\hat{R}_{\mathcal{N}}$ prefers different operators in terms of trainability and Expressivity. Their combination can filter out bad architectures in both aspects and strongly correlate with networks' final performance.

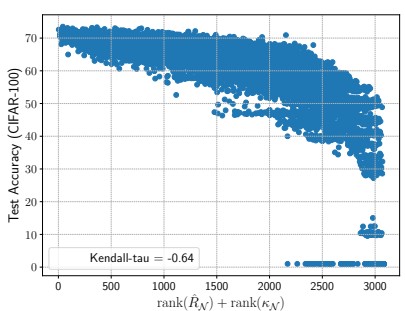

**Figure 9:** Summation of ranking of $\kappa_{\mathcal{N}}$ and $\hat{R}_{\mathcal{N}}$ exhibits stronger (negative) correlation with the test accuracy of architectures in NAS-Bench201 (Dong & Yang, 2020).

## E Generalization v.s. Test Accuracy

Conceptually, the generalization gap is the difference between a model's performance on training data and its performance on unseen data drawn from the same distribution (e.g., testing set). In comparison, the two indicators $\kappa_{\mathcal{N}}$ (trainability) and $R_{\mathcal{N}}$ (expressiveness) of a network determine how well the training set could be fit (i.e., training set accuracy), and do not directly indicate its generalization gap (or test set accuracy). Indeed, probing generalization of an untrained network at its initialization is a daunting, open challenge that seems to go beyond the current theory scope.

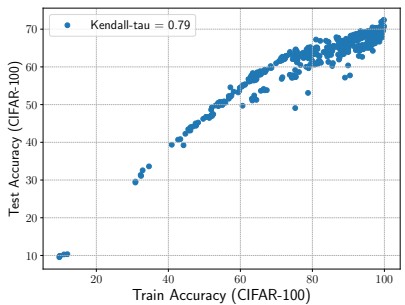

**Figure 10:** The correlation between test accuracy and the training accuracy in NAS-Bench201 (Dong & Yang, 2020).

In NAS, we are searching for the architecture with the best test accuracy. As shown in Figure 10, in NAS-Bench201 the training accuracy strongly correlates with test accuracy. This also seems to be a result of the current standard search space design that could have implicitly excluded severe overfitting. This explains why $\kappa_{\mathcal{N}}$ and $R_{\mathcal{N}}$, which only focuses on trainability and expressiveness during training, can still achieve good search results of test accuracy.

