# OpenReview forum: "Neural Architecture Search on ImageNet in Four GPU Hours: A Theoretically Inspired Perspective"
_ICLR.cc/2021/Conference — ICLR 2021 Poster_

### Official Review · AnonReviewer1 · 2020-10-27
**Interesting idea, but more analysis is needed to justify the idea**

**Rating:** 6
**Confidence:** 4

**Review:**

# Summary
This paper aims to speed up NAS with a training-free performance estimation strategy, i.e., estimate the performance of an architecture at initialization without training (not necessarily the absolute performance, but the relative ranking of architectures).

The proposed strategy estimates an architecture’s performance from two perspectives: (1) trainability, and (2) expressivity. The metrics to measure trainability and expressivity are inspired by recent progress on deep learning theory. Specifically, The trainability of an architecture is evaluated by the spectrum of its NTKs, and the expressivity is evaluated by the number of linear regions in its input space.

The authors also propose a pruning-based search algorithm that relies on the proposed performance estimation strategy. Experiments on two search spaces NAS-Bench-210 and DARTS show that the proposed method can find architectures with reasonable performance while being much faster.

# Strong points
1. Training-free NAS is conceptually novel and an interesting direction. This could make NAS more accessible and useful in practice if training-free methods can work.
2. The authors draw inspiration from recent deep learning theory to design the training-free performance estimation metric.
3. The proposed NAS method can complete the search much faster than previous NAS methods.

# Weak points
1. The metrics for trainability and expressivity seem to be the direct application/extension of recent deep learning theories (NTK, linear region). This makes the novelty of this work a bit weak.
2. It’s unclear how the number of linear regions is computed. The authors gave a definition in Equation 4 but didn’t mention how exactly the number of linear regions is computed.
3. The Kendall tau correlation between NTK/linear region and the test accuracy is actually weak based on the reported value (-0.42 / 0.5). Why would this weak correlation lead to an architecture with good performance? Also, the final method uses a combined relative ranking. What’s the Kendall tau for this combined relative ranking metric?
4. The most interesting part in this work is the proposed training-free metric. It is important to compare this training-free metric with the typical metric (e.g., validation accuracy after training a limited number of epochs), while keeping other factors unchanged. What will the performance/search efficiency be if we combine the proposed metric with existing search algorithms (e.g., random search, reinforcement learning)? It will be helpful to compare the proposed metric and the typical metric using the same search algorithm and the same number of samples.
5. It’s unclear how important the pruning mechanism is for the final performance/search efficiency. Having the above analysis/results will also help answer this question.
6. The absolute performance on ImageNet in DARTS search space is not very strong. Also, for experiments on NAS-Bench-201, the authors only compare to some relatively weak baselines. As shown in Table 5 in the original NAS-Bench-201 paper, random search/REINFORCE perform on par or better than the proposed method, especially on ImageNet-16-120.

# Justification of rating
The proposed metric is interesting. My main concern is the lack of analysis as mentioned above. Without those analyses, it’s hard to convince people that this metric is empirically useful. Also, the novelty is a bit limited as the metric seems to be a direct application/extension of recent deep learning theories.



# Additional feedback

“Training a supernet till convergence is extremely slow” - This sentence is not very accurate. Training a supernet is usually similar to training one architecture on the dataset. Although it’s more expensive than training-free, it’s practically acceptable. The drawback of supernet is more about its inaccurate performance estimation.

“What to select” sounds more like the search space definition in NAS. The objective of NAS is always clear (best accuracy under certain FLOPS/params). I suggest the authors rephrase the text.

Figure 5. The two figures have some overlappings.

# After rebuttal
I would like to thank the authors for their extensive efforts during the rebuttal. My main concerns are resolved, so I change the rating to borderline accept. I encourage the authors to update the paper with the results provided in the rebuttal, especially the explanation of the novelty and the results for "Proposed metric with existing search algorithms".

---

> ### Author Response · Authors · 2020-11-20
> **Response to Reviewer1 (part 3/3): more ablation studies. [cont’d]**
>
> * **Performance on ImageNet**
>
> First, we emphasize that the main pursuit of our work is ultra-fast and low-cost search, at which our work seems to be “unbeatable”. On CIFAR-10, our search time (0.02 GPU day) outperforms the closest prior work, PC-DARS [1], over 5$\times$. On ImageNet, if we search on ImageNet (with mark $\dagger$), we outperform PC-DARS [1] over 22.35$\times$.
>
> Second, the latest numbers in the last row of Table 3, using the standard training recipe, are now updated to 24.5% top-1 and 7.5% top-5, both on par with PC-DART searches on ImageNet, yet using 22.35$\times$ less search time. Our training of ImageNet-searched architecture was not fully completed before the ICLR deadline, and we continued to train it until convergence after our submission. That confirms the effectiveness of our training-free NAS that is competitive to existing NAS algorithms, yet still being extremely efficient. NAS-Bench-201 is a popular testbed where TE-NAS is clearly superior in both performance and efficiency.
>
> We also suspect a layout issue in the original draft Table 3 might cause some misinterpretation of our significant results. Table 3 includes results searched on CIFAR-10 and transferred to ImageNet, and directly searched on ImageNet. The two categories are not directly comparable, and the latter will perform better but cost longer. The mark $\dagger$ indicates architectures searched on ImageNet, otherwise it is searched on CIFAR-10 or CIFAR-100. We have revised our Table 3 layout to make the display more clear.
>
> We firmly confirm the release of all our codes and searched architectures to ensure the reproducibility of our results.
>
> [1] Xu, Yuhui, et al. "Pc-darts: Partial channel connections for memory-efficient differentiable architecture search." ICLR 2020.
>
> * **SOTA on NAS-Bench201**
>
> The “REA/RS/REINFORCE/BOHB” methods used the meta-data (i.e. ground truth accuracy) provided by the NAS-Bench201 database. This is the reason why they achieve higher accuracy while only costing a few seconds. Our selected baselines are the state-of-the-art ones in the fair setting. We privately double-confirmed this with the author of NAS-Bench201.
>
> * **Slow training of supernet**
>
> Thank you for your question. Regarding the slow training of cell-based networks (each cell is a directed acyclic graph of many operators, e.g. the supernet for DARTs or NAS-Bench-201  search space), we have the following two folds of observations:
> 1) On CIFAR-100, the architecture in NAS-Bench-201 with the highest number of parameters: 1.47MB parameters, and costs 0.2G FLOPs.
> However, for ResNet18: 11.1MB parameters, only 35.9MB FLOPs.
> The reason is that cell-based architecture suffers from repeated merges (adds) of feature maps.
> 2) The slowness of supernet training is also discussed in the community:
>
>   (a) https://github.com/quark0/darts/issues/80#issuecomment-503075338
>
>   (b) https://github.com/quark0/darts/issues/37#issuecomment-417390963
>
>
>
> * **More language and figure issues**
> Thank you for your great suggestion and we happily take your suggestion to improve the clarification. We have rephrased "what to select" all with “how to evaluate” in our draft. We have also modified our Figure 5.

---

> ### Author Response · Authors · 2020-11-20
> **Response to Reviewer1 (part 2/3): more ablation studies.**
>
> * **How the number of linear regions is computed**
>
> As mentioned in section 3.1.2 (after Eq. 4), we calculated the number of linear regions equal to the number of unique activation patterns produced by a network. Therefore, the computation steps are:
>
> 1) for each input image, collect features output by all ReLU layers in the network;
>
> 2) concatenate and transform these features into a binary code (1 for positive value and 0 otherwise);
>
> 3) calculate how many unique binary code given the training data.
>
> We will release all experiment codes including this computational tool, after paper acceptance.
>
>
> * **Combined relative ranking**
>
> Since two indicators have different preference over operators (Table 5), their combination exhibits more powerful selection capability over the search space. In Figure 9 in section D3, we show that the correlation between test accuracy and the summation of two rankings is much higher (0.64, ranking the lower the better).
>
>
> * **Proposed metric with existing search algorithms**
>
> On CIFAR-100, we conduct the following experiments:
> 1) by using validation accuracy as selection metric (each sampled architecture is trained by one epoch, i.e. proxy inference):
>
>   (a) Random: 66.99 (2.89) by sampling 477 architectures, spent 581 min
>
>   (b) RL: 69.29 (1.51) by sampling 441 architectures, spent 541 min
>
>   (c) Evolution: 69.36 (1.36) by sampling 426 architectures, spent 516 min
>
> 2) by using $\kappa$ and the number of linear regions:
>
>   (a) Random: 67.71 (0.73) by sampling 498 architectures, spent 62 min
>
>   (b) RL: 70.88 (0.60) by sampling 492.75 architectures, spent 61 min
>
>   (c) Evolution: 70.84 (1.26) by sampling 276 architectures, spent 69 min
>
> We can conclude that by using our training free indicators, all three sampling based NAS methods can achieve higher performance with drastically reduced search time costs.
>
> The pruning mechanism can further boost the search efficiency of our already lightweight framework. Sampling-based methods suffer from time costs that scale up with the complexity of search space. They all have to sample a (large) number of architectures before converging,  As we explained in Section 3.2, our pruning-by-importance mechanism quickly shrinks the search possibilities from the beginning, without any training. The proposed pruning way can reduce the sampling cost during the search from exponential to polynomial. This means sampling-based NAS methods are intrinsically more costly than our pruning-based way, even equipped with training-free indicators.

---

> ### Author Response · Authors · 2020-11-20
> **Response to Reviewer1 (part 1/3): our contribution is unique and strong, and requires non-trivial efforts.**
>
> We thank you for the comment, but respectfully argue that putting theory tools in the NAS application requires non-trivial efforts.
>
> Our main and foremost contribution, i.e., bridging the gap between existing deep learning and NAS application, is neither easy nor straightforward. For one example, many theory papers study simple MLPs for example, and some assume (unrealistically) ultra-wide or ultra-deep networks. It remains to be verified how those theories or measures work on quantitatively assessing diverse practical networks, with various connectivity, normalization, branching and so.
>
> Further noteworthy is that there has been no prior work discussing whether one can reliably rank architectures based on theory-based indicators. Especially in a large NAS search space, there are massive candidate architectures, and every architecture has its similar neighbors (e.g, by just mutating an operator or a path). Given both the high space complexity and the inter-architecture similarity, it was really uncertain to anyone before, whether theory-based indicators can precisely characterize each architecture w.r.t. Its neighbors, and can effectively rank them to find the top performers. Our work gives a decisive empirical answer for the first time.
>
> Moreover, extending those theory-based indicators to NAS sees an important challenge of efficiency. For example, on NAS-Bench-201, there are >15k networks to be evaluated; meanwhile, computing linear regions as [1,2] described are extremely time-consuming. To scale up efficiently, we discard vanilla hashing of ReLU activation patterns and formulate the problem as bitwise operations and matrix multiplication.
>
> Lastly, we emphasize the most important and novel contribution of this work is to present a brand new view to rethink NAS. We quote AnonReviewer4 who has precisely summarized our contributions: “This work bridges between the advances in deep learning theory and the practice of NAS. I think it can become an important starting point towards theoretically inspired NAS and can motivate the discovery of more practically viable deep network performance indicators”. More concretely:
> * Almost all previous NAS works stick to one measure for ranking and search architectures: the validation loss/accuracy of a sampled architecture. However, this is known to be invariably flawed, e.g., due to the truncated training [3,4] or unfair comparison under different training hyperparameters [4,5].
> * We are the first to view NAS, or more generally, neural network model selection, from two disentangled perspectives trainability and expressiveness (different preference of k and R in Table 5). This opens up the black box of validation performance and enables us to dive much deeper into understanding the search behaviors. For example, our Figure 5 provides an interpretable search trajectory of what NAS algorithms respectively emphasize, in the early and later search stages.  As AnonReviewer4 nicely commented: “It is very interesting to see that the supernet first is pruned by quickly increasing the network’s trainability, and then fine-tuning the expressivity in a small range.”
> * Lastly but importantly, the organic utilization of two theory-based indicators has contributed to a new scope of training-free NAS, which all reviewers appreciate as “promising” ”conceptually novel and an interesting”. Our framework searches for the same competitive models under extremely low costs that no previous framework was able to match, which we proudly highlighted in our title.
>
> We sincerely hope the reviewer will take them into account, and make a more fair and positive reassessment of our work, which we would sincerely appreciate.
>
> [1] Hanin, Boris, and David Rolnick. "Complexity of linear regions in deep networks." Neurips 2019.
>
> [2] Xiong, Huan, et al. "On the Number of Linear Regions of Convolutional Neural Networks." ICML 2020.
>
> [3] Zheng, Xiawu, et al. "Multinomial distribution learning for effective neural architecture search." CVPR 2019.
>
> [4] Yang, Antoine, et al. "NAS evaluation is frustratingly hard." ICLR 2020.
>
> [5] Dong, Xuanyi, et al. "AutoHAS: Efficient hyperparameter and architecture search." arXiv preprint arXiv:2006.03656 (2020).

---

### Official Review · AnonReviewer4 · 2020-10-27
**Paper 2426 Review**

**Rating:** 8
**Confidence:** 4

**Review:**

Summary: Training-free NAS is a promising direction. This work demonstrates that two theoretically inspired indicators: the spectrum of NTK and number of linear regions, strongly correlate with the network’s performance, and can be leveraged to reduce the search cost and decouple the analysis of the network’s trainability and expressivity. The authors thus proposed TE-NAS to rank architectures by analyzing the two indicators. Without involving any training, TE-NAS can achieve competitive NAS performance within minimum search time.

Pros:

+ This work bridges between the advances in deep learning theory and the practice of NAS. I think it can become an important starting point towards theoretically inspired NAS and can motivate the discovery of more practically viable deep network performance indicators.

+ The authors made efforts to add interpretability to their method and to understand the search process, e.g. Figure 4 and section 3.2.1. It is very interesting to see that the supernet first is pruned by quickly increasing the network’s trainability, and then fine-tuning the expressivity in a small range.

+ On NAS-Bench-201 and DARTS search spaces, the propose method’s results are extremely promising. For the latter space, the proposed method can search within 30 minutes (on CIFAR-10) and 4 hours (on ImageNet), while the search models’ accuracy remain among the most competitive few.

Cons:

- I remain skeptical how much the initial stage expressiveness/trainability would last and be preserved until end of training. Also, the two do not represent the full spectrum of desirable model characteristics. The ultimate goal of NAS is to seek a model of best generalization ability, which cannot be merely or easily from its expressiveness (fitting ability, i.e., achievable training error) and trainability (optimization difficulty).

- Although the authors claimed that the two chosen indicators are strongly correlated with the network’s test accuracies, I’m not sure how true that actually is. In fact, probing the generalization from a trained model is already challenging, not to say from an untrained one. The authors might need to discuss more or to tone down further.

- Figures 1 & 2 leave me with impression that both indicators can filter out some bad outlier architectures; but for many relatively good accuracy architectures, the correlations between the two and the true accuracy are not necessarily high (overall Kendall-tau magnitudes are between 0.4 and 0.5).

- The authors also didn’t specify the computational complexity of computing the two indicators (average per architecture), except just saying they’re very fast.


- More NAS works are related to the manuscript, for example [1][2][3].
[1]SGAS: Sequential Greedy Architecture Search
[2] GP-NAS: Gaussian Process Based Neural Architecture Search
[3] HourNAS: Extremely Fast Neural Architecture Search Through an Hourglass Lens

- (minor) The writing quality is in general good; some occasional typos:
training-free AS framework: “AS” -> “NAS”;
no only -> not only

---

> ### Author Response · Authors · 2020-11-20
> **Response to Reviewer4: Thank you for your great questions and suggestions!**
>
> We greatly appreciate your questions and suggestions, and we really like them! Below are our responses.
>
> 1. "initial stage expressiveness/trainability"
>
> Your question is very much to-the-point and we like it a lot. Actually, we also started investigating similar questions right after this submission.
>
> We recently observed some trends of the NTK spectrum during training shared by different networks. For example, in early training (e.g. within the first epoch) $\kappa$ first increase then drops. We find this observation matches the recent studies on the evolution of the network's Hessian spectrum during training [1], where large negative eigenvalues quickly disappear after the early iterations. We are actively working on this open problem.
>
> Meanwhile, in our submission, we demonstrate that $\kappa$ and R at initialization can already filter-out bad architectures, which is very effective and helpful for NAS search. Of course, more in-depth analysis of these two indicators at the later stages of NAS are of our vital interest.
>
> 2. About Generalization
>
> This is really an excellent and insightful comment.
>
> In a general case, it is true that only $\kappa$ (trainability) and $R$ (expressiveness) do not add up to the full picture of generalization. Conceptually, the generalization gap is the difference between a model’s performance on training data and its performance on unseen data drawn from the same distribution (e.g., testing set). In comparison, the two indicators of trainability and expressiveness for a network basically determine how well the training set could be fit (i.e., training set accuracy), and do not directly indicate its generalization gap (or test set accuracy). Indeed, probing the generalization of an untrained network at its initialization is a daunting, open challenge that seems to go beyond the current theory scope.
>
> Yet very interestingly, NAS might happen to be such a special case that $\kappa$ and $R$ together indeed work effectively. The goal of NAS, practically, is to find the architecture from the search space with the high test accuracy on the target test set. We show the relationship between test and training accuracies in section E in our new supplement. As we can see from Figure 10, at least on CIFAR-100 networks achieving high test accuracy also tend to achieve high training accuracy. This seems to be a result of the current standard search space design, which could have implicitly excluded severe overfitting. This explains why in our submission, only focusing on trainability and expressiveness can still achieve good search results of test accuracy.
>
> Thank you for pointing out! We will add the above discussion in the revised draft and will properly tune our tone.
>
> 3. About Kendall-tau correlations
>
> Since two indicators have different preferences over operators (Table 5), their combination exhibits more powerful selection capability over the search space. In Figure 9 in section D3 we show that the correlation between test accuracy and the summation of two rankings is much higher (0.64, ranking the lower the better).
>
> 4. About computation complexity
>
> Besides the search time costs reported in Table 1~3, here we measure the FLOPs of calculating $\kappa$ and the number of linear regions on CIFAR-100.
>
> In comparison with the FLOPs of proxy inference (e.g. train the sampled architecture for one epoch and approximate its validation accuracy), our $\kappa$ and $R$ only cost 0.06% and 0.03% FLOPs, respectively. However, one-epoch proxy inference only achieves 0.132 kendall-tau correlation by our experiments, much lower than our 0.64.
>
> 5. Typos and missing references
>
> Thank you! We have added the missing references and fixed the typos.
>
> [1] Ghorbani B, Krishnan S, Xiao Y. An investigation into neural net optimization via hessian eigenvalue density. arXiv preprint arXiv:1901.10159. 2019 Jan 29.

---

> > ### Comment · AnonReviewer4 · 2020-11-23
> > **Thanks for your response! This is a great paper.**
> >
> > I appreciate the authors for their extensive clarifications, to me and for other reviewers. Reading them has cleared my previous concerns.
> >
> > Particularly interesting is the discussion about generalization: the prediction of expressiveness/trainability become the "surrogate" task for estimating the searched model's generalization/testing performance, because training and test accuracies correlate well in the current NAS space. It is an interesting insight.
> >
> > While I agree other reviewers this paper mainly exploits DL theory tools invented before, I side with the authors that, cleverly using DL theory tools - for the first time - in NAS is a very novel and non-trivial contribution alone. The efficient search by pruning framework "without training" is also smart, quickly shrinking possibilities from the beginning. The accuracy results are competitive after the authors clarified some settings.
> >
> > In my view, this is a strong submission well deserving ICLR acceptance. I raise my rating from 7 to 8.

---

### Official Review · AnonReviewer2 · 2020-10-28
**Well written paper but incremental improvements.**

**Rating:** 6
**Confidence:** 4

**Review:**

This paper introduces a searching framework of neural architectures ranking the candidates with two different metrics: the spectrum of NTK and the number of linear regions in the input space. These two metrics do not require the training of neural networks lightening the computational burdensome. Authors support their method by providing results from CIFAR-10, ImageNet, and NAS-Bench-201 benchmark (Dong & Yang).

Overall, this paper proposes new metrics that can be used in the NAS search. Despite the paper well written, the performance of their methodology is incremental improvements (or par) among various existing NAS algorithms. I would recommend a marginally above acceptance threshold for now.

Strength
1. The author provides two new different metrics: a condition number of NTK and cardinality of linear regions to select good architectures besides the validation loss.
2. Experiments not only on CIFAR10, ImageNet, but also NAS-Bench 201
3. Free from the training allows their method to free from computational burdensome allowing to stack an equivalent number of cells for both the search and evaluation phase.

Weakness
1. The experiment results are par (or worse) to the existing NAS algorithms. By listing the recently published NAS literatures on CIFAR-10: P-DARTs: 2.50%, DATA: 2.59%, SGAS: 2.66%, PC-DARTs: 2.57%, RandomNAS-NSAS: 2.64%. Especially, SGAS and PC-DARTs require 0.25 and 0.1 GPU days respectively. (However, NAS-BENCH-201 results are competitive.)
2. While no training has a benefit in the computational budget, the performance could be worse than the method with training (less information) as listed in Weakness 1.

Questions
1. According to Definition 1, the number of linear regions depends on the fixed set of parameters $\theta$. How much the number of linear regions affected by the $\theta$? What happens you use the trained $\theta$ or draw the network parameters with other initializing methods such as Ha initialization or Xavier initialization?
2. Figure 5 shows the pruning trajectory based on the author's method. Intuitively randomly pruning the operation will also reduce the $\kappa$ and the linear regions. Can you support the pruning efficiency based on the proposed metrics by comparing it with the random pruning?
3. Have you tried weight sum loss such as $\alpha \Delta \kappa + (1-\alpha) \Delta R$ where $\alpha \in (0, 1)$ be a hyperparameter? Table 4 seems $R$ plays more significant role than $\kappa$.

Reference:
1. Xu, Y et al. (2019, September). PC-DARTS: Partial Channel Connections for Memory-Efficient Architecture Search. ICLR2020
2. Chen, X et al. Progressive differentiable architecture search: Bridging the depth gap between search and evaluation. ICCV2019
3. Chang, J et al. DATA: Differentiable ArchiTecture Approximation. Neurips2019
4. Li, G et al. SGAS: Sequential Greedy Architecture Search. CVPR2020
5. Zhang, M et al. Overcoming Multi-Model Forgetting in One-Shot NAS with Diversity Maximization. CVPR2020
6. Dong & Yang. NAS-Bench-201: Extending the Scope of Reproducible Neural Architecture Search. ICLR 2020

---

> ### Author Response · Authors · 2020-11-20
> **Response to Reviewer2 (part 2/2): more ablation studies.**
>
> 1. Thank you for your great question! In summary, different initializations do have some effects on the number of linear regions, and we are also interested in studying this on real-world networks. We chose one architecture from NAS-Bench-201 and compared the number of linear regions with different initializations over four random draws (on CIFAR-100). Gaussian: 3651.0 (37.6); Kaiming_uniform: 3062.8 (318.2); Xavier_uniform: 2681.3 (320.8). We can see the number of linear regions measured under Gaussian initialization benefits from the smallest variance, which contributes to the stability of our NAS search.
>
> 2. This is a really great suggestion. We conducted random pruning and recorded the trajectory of the supernet (each measure is averaged over three times):
> $R$ (higher the better): 1100.7 $\rightarrow$ 1098.3 $\rightarrow$ 1022.7 $\rightarrow$ 544.7
> $\kappa$ (lower the better): 246.5 $\rightarrow$ 220.1 $\rightarrow$ 399.8 $\rightarrow$ 305.5
> We can see random pruning may potentially drop the number of linear regions, but not necessarily always reduce $\kappa$ since it may blindly prune some useful operators while keeping bad ones. This random pruning ends with only 43.35% test accuracy on CIFAR-100.
>
> 3. As we indicated on page 5 footnote, We tried some weighted summations of the two, and found their equal-weight summation to perform the best. We conducted your suggested convex combination study with $\alpha$ from 0 ~ 1, and the best result was 70.54 (0.71) achieved with $\alpha$ = 0.6.

---

> ### Author Response · Authors · 2020-11-20
> **Response to Reviewer2 (part 1/2): our final accuracy is on par or even better, however, we achieve extreme search efficiency.**
>
> Thank you for your question!
>
> First, we emphasize that the main pursuit of our work is ultra-fast and low-cost search, at which our work seems to be “unbeatable”. On CIFAR-10, our search time (0.02 GPU day) outperforms the closest prior work, PC-DARS [1], over 5$\times$. On ImageNet, if we search on ImageNet (with mark $\dagger$), we outperform PC-DARS [1] over 22.35$\times$.
>
> Second, the latest numbers in the last row of Table 3, using the standard training recipe, are now updated to 24.5% top-1 and 7.5% top-5, both on par with PC-DART searches on ImageNet, yet using 22.35$\times$ less search time. Our training of ImageNet-searched architecture was not fully completed before the ICLR deadline, and we continued to train it until convergence after our submission. That confirms the effectiveness of our training-free NAS that is competitive to existing NAS algorithms, yet still being extremely efficient. NAS-Bench-201 is a popular testbed where TE-NAS is clearly superior in both performance and efficiency.
>
>   * We suspect a layout issue in the original draft Table 3 might cause some misinterpretation of our significant results. Table 3 includes results searched on CIFAR-10 and transferred to ImageNet, and directly searched on ImageNet. The two categories are not directly comparable, and the latter will perform better but cost longer. The mark $\dagger$ indicates architectures searched on ImageNet, otherwise it is searched on CIFAR-10 or CIFAR-100. We have revised our Table 3 layout to make the display more clear.
>
> We firmly confirm the release of all our codes and searched architectures to ensure the reproducibility of our results.
>
> We hope the above clarifications could turn your assessment into a more positive one, which we will sincerely appreciate.

---

### Official Review · AnonReviewer3 · 2020-10-28
**Review Comment**

**Rating:** 4
**Confidence:** 5

**Review:**

This work studied an interesting topic of training-free Neural Architecture Search (NAS). It utilizes two training-free indicators to measure the performance of a network without training it. The experiments show the proposed approach has improvement in searching time.

However,  as the main contribution, the two measurements of the training-free indicators (trainability and expressivity) already be proposed by previous works. The rest contributions of this work are the proposed NAS networks, Instead of using the original loss function to guide the search process, this work simply combined those two measurements as a new ranking identifier of candidate architectures. The contributions not strong enough.

Advantages:
1. The writing quality of the paper is good enough
2. The paper has good descriptions of the related work.

Key weakness:
1. The major contributions of this work, the two network measurement methods, were proposed by previous works.
The first indicator:
    Arthur Jacot, Franck Gabriel, and Clement Hongler. Neural tangent kernel: Convergence and generalization in neural networks. In Advances in Neural Information Processing Systems 31. 2018a.
    Jaehoon Lee, Lechao Xiao, Samuel Schoenholz, Yasaman Bahri, Roman Novak, Jascha SohlDickstein, and Jeffrey Pennington. Wide neural networks of any depth evolve as linear models under gradient descent. In Advances in neural information processing systems, pp. 8572–8583, 2019.
    Yeonjong Shin and George Em Karniadakis. Trainability of relu networks and data-dependent initialization. Journal of Machine Learning for Modeling and Computing, 1(1), 2020.
    Lenaic Chizat, Edouard Oyallon, and Francis Bach. On lazy training in differentiable programming. 2019.

The seconde indicator:
    Huan Xiong, Lei Huang, Mengyang Yu, Li Liu, Fan Zhu, and Ling Shao. On the number of linear regions of convolutional neural networks. arXiv preprint arXiv:2006.00978, 2020.

2. The differences between the proposed pruning-based NAS and previous work are not clear. What is the key novelty of the proposed pruning strategy?
3. According to Table 3, it seems that the TE-NAS couldn’t find the optimal neural architecture like P-DARTS and PC-DARTS, which confirms the limitation of this training-free search framework.

---

> ### Author Response · Authors · 2020-11-20
> **Response to Reviewer3 (part 3/3): our final accuracy is on par or even better, however, we achieve extreme search efficiency.**
>
> Thank you for your great question, although we cannot agree it “confirms the limitation of this training-free search framework.”
>
> First and foremost, we emphasize again that the main pursuit of our work is ultra-fast and low-cost search, at which aspect our work seems to be “unbeatable”. On CIFAR-10, our search time outperforms the closet prior work, PC-DARS [1], over 5$\times$. On ImageNet, if we search on ImageNet (with mark "$\dagger$"), we outperform PC-DARS [1] over 22.35$\times$.
>
>   * We suspect a layout issue in the original draft Table 3 might cause some misinterpretation of our significant results. Table 3 includes results searched on CIFAR-10 and transferred to ImageNet, and directly searched on ImageNet. The two categories are not directly comparable, and the latter will perform better but cost longer. The mark “†” The architecture is searched on ImageNet, otherwise it is searched on CIFAR-10 or CIFAR-100. We have revised our Table 3 layout to make the display more clear.
>
> Second, our training of ImageNet-searched architecture was not fully completed before the ICLR deadline, and we actually continued to train that until convergence after our submission. The latest numbers in the last row of Table 3, using the standard training recipe, now become 24.5% top-1 and 7.5% top-5, both on par with PC-DART searches on ImageNet, yet still using 22.35$\times$ less search time. That confirms the non-compromised effectiveness of our training-free NAS, besides being extremely efficient.
> We firmly confirm the release of all our codes, searched architectures and pre-trained weights to ensure the reproducibility of our results.
>
> [1] Xu, Yuhui, et al. "Pc-darts: Partial channel connections for memory-efficient differentiable architecture search." ICLR 2020.

---

> ### Author Response · Authors · 2020-11-20
> **Response to Reviewer3 (part 2/3): our pruning-based search is more efficient.**
>
> The goal of pruning-based NAS is to further accelerate the search efficiency of our already lightweight framework. As we explained in Section 3.2, the pruning-by-importance mechanism quickly shrinks the search possibilities from the beginning, without any training. A similar pruning-from-scratch style was recently explored in-network compression (Lee et al., 2018; Wang et al., 2020). Our pruning method can reduce the sampling cost during the search from exponential to polynomial. This means sampling-based NAS methods are intrinsically more costly than our pruning-based way, even equipped with training-free indicators (please refer to our response to R1 about results of RL/Evolution methods with our $\kappa$ and $R$).
>
> Previous pruning-related NAS works [1,2,3] mainly prune the supernet, i.e., removing bad operators like zeros, pooling, skip-connect, etc. They still have to meanwhile train the supernet, and use the loss gradient signals to choose over the remaining operators (1$\times$1, 3$\times$3 convolutions). In contrast, our pruning-based framework is built with theory-inspired, training-free indicators. Since no training is involved, there is no alternating training and pruning as [1] did. Instead, we could directly prune the full supernet into a single-path network, as outlined in Algorithm 1. Obviously, this framework is also fundamentally novel and different from any existing NAS work.
>
> [1] Xu, Yuhui, et al. "Pc-darts: Partial channel connections for memory-efficient differentiable architecture search." ICLR 2020.
>
> [2] Luo, Renqian, et al. "Neural architecture search with gbdt." arXiv preprint arXiv:2007.04785 (2020).
>
> [3] Dai, Xiyang, et al. "DA-NAS: Data Adapted Pruning for Efficient Neural Architecture Search." arXiv preprint arXiv:2003.12563 (2020).

---

> ### Author Response · Authors · 2020-11-20
> **Response to Reviewer3 (part 1/3): our main contribution is unique and strong.**
>
> With our due respect for the reviewer’s time, we humbly yet firmly suggest that the reviewer might have an important misunderstanding or underestimation of this paper’s true merit.
>
> We start by strongly pointing out: “The major contributions of this work, the two network measurement methods” is an incorrect assessment of our main points. We quote Reviewer4 who precisely summarized our contributions: “This work bridges between the advances in deep learning theory and the practice of NAS. It can become an important starting point towards theoretically inspired NAS and can motivate the discovery of more practically viable deep network performance indicators”. We itemize our responses below:
> 1. The theory references and why they do not compromise our contribution
>   * We appreciate your references. The authors are experienced researchers in the deep learning theory and NTK field. All the mentioned references are familiar to us, and we have already cited all of them in our submission, if you could please check our reference list again
>   * However, we are not intended to be yet another NTK theory work, and we intend not to overemphasize the theoy background to distract our target audience. Instead, we very clearly poised our unique position: to show the NAS audience the power and potential of leveraging the latest deep network theories. We are happy to see our goal well-received by other reviewers
> 2. Putting theories into NAS application requires non-trivial efforts
>   * Our main and foremost contribution, i.e., bridging the gap between existing deep learning and NAS application, is neither easy nor straightforward. For example, many theory papers study simple MLPs, and some assume (unrealistically) ultra-wide or ultra-deep networks. It remains to be verified of how these theories quantitatively work on assessing diverse practical networks, with various connectivity, normalization, branching, etc.
>   * Further noteworthy is that there has been no prior work discussing whether one can reliably rank architectures based on theory-based indicators. Especially in a large NAS search space, there are massive candidate architectures, and each has its similar neighbors (e.g, by just mutating an operator or a path). Given both the high space complexity and the inter-architecture similarity, it was really uncertain to anyone before, whether theory-based indicators can precisely characterize each architecture w.r.t. its neighbors and effectively rank them to find top performers. We give a decisive empirical answer for the first time
>   * Moreover, extending theory-based indicators to NAS sees an important challenge of efficiency. For example, on NAS-Bench-201, there are >15k networks to be evaluated; meanwhile, computing linear regions as [1,2] described is extremely time-consuming. To scale up efficiently, we discard vanilla hashing of ReLU activations and formulate the problem as bitwise operations and matrix multiplication
> 3. We present a brand new view to rethink NAS
>   * Almost all previous NAS works stick to one measure for ranking and search architectures: the validation loss/accuracy of a sampled architecture. However, this is known to be invariably flawed, e.g., due to the truncated training [3,4] or unfair comparison under different hyperparameters [4,5]
>   * We are the first to view NAS, or more generally, neural network selection, from two disentangled perspectives trainability and expressiveness (different preference of $\kappa$ and $R$ in Table 5). This opens up the black box of validation performance and enables us to dive much deeper into understanding the search behaviors. Our Figure 5 provides an interpretable search trajectory of what NAS algorithms respectively emphasize, in the early and later search stages.  As Reviewer4 commented: “It is very interesting to see that the supernet first is pruned by quickly increasing the network’s trainability, and then fine-tuning the expressivity in a small range.”
>   * Lastly but importantly, the organic utilization of two theory-based indicators has contributed to a new scope of training-free NAS, which all reviewers appreciate as “promising” ”conceptually novel and an interesting”. Our framework searches for the same competitive models under extreme low costs that no previous framework was able to match, which we proudly highlighted in our title
>
> We hope the above explanations have clarified any misunderstanding or confusion. We sincerely hope the reviewer takes them in account and makes a more fair and positive reassessment of our work in the correct context.
>
> [1] Hanin et al. Complexity of linear regions in deep networks. Neurips19
>
> [2] Xiong et al. On the Number of Linear Regions of Convolutional Neural Networks. ICML20
>
> [3] Zheng et al. Multinomial distribution learning for effective neural architecture search. CVPR19
>
> [4] Yang et al. NAS evaluation is frustratingly hard. ICLR20
>
> [5] Dong et al. AutoHAS: Efficient hyperparameter and architecture search. 2020

---

### Public Comment · ~Thiago_Serra1 · 2020-11-12
**Background on linear regions**

I really like the idea of using the number of linear regions as a metric for NAS as the authors have proposed in this paper.

However, I think that the authors should properly cite prior work on the topic: only two recent papers on linear regions are included. There are at least three prior publications using the concept of activation patterns as a means to characterize the linear regions of neural networks:

[1] http://proceedings.mlr.press/v70/raghu17a/raghu17a.pdf

[2] https://www.researchgate.net/publication/322539221_Notes_on_the_number_of_linear_regions_of_deep_neural_networks

[3] http://proceedings.mlr.press/v80/serra18b/serra18b.pdf

Among those, the credit for first use rightfully goes to [1]. But given the importance of linear regions for your work, I believe that you should consider including a thorough literature review on linear regions.

---

> ### Author Response · Authors · 2020-11-20
> **Thank you very much for your important references!**
>
> Thank you very much for your great suggestions! We have added these important references in our submission and updated our related works about linear regions in our submission.

---

### Decision · Program_Chairs · 2021-01-07
**Final Decision**

**Decision:**

Accept (Poster)

**Comment:**

The authors propose training-free neural architecture search using two theoretically inspired heuristics: the condition number of the Neural Tangent Kernel (to measure "trainability" of the architecture), and the number of linear regions in the input space (to measure "expressivity"). These two heuristics are negatively and positively correlated with test accuracy, respectively, allowing for fast, training-free Neural Architecture Search. It is certainly not the first training-free NAS proposal, but achieves competitive results with much more expensive NAS methods.

A few reviewers mentioned limited novelty of the method, a claim with which I agree. The contribution of the paper, however, is something different than how it was presented. The core message seems to be that the two proposed heuristics can greatly speed up NAS, and should be a baseline method against which more expensive methods should test.

I feel like this is a borderline paper, but may be of interest to researchers in the field.